# Combinatorial ERK Inhibition Enhances MAPK Pathway Suppression in BRAF-Mutant Melanoma

**DOI:** 10.3390/ijms26199794

**Published:** 2025-10-08

**Authors:** Corinna Kosnopfel, Tobias Sinnberg, Shrunal Mane, Michelle Dongo, Claus Garbe, Heike Niessner

**Affiliations:** 1Department of Hematology, Oncology and Pneumology, University Hospital Muenster, 48149 Münster, Germany; 2Division of Dermatooncology, Department of Dermatology, University of Tuebingen, 72076 Tuebingen, Germany; tobias.sinnberg@med.uni-tuebingen.de (T.S.); shrunal.mane@med.uni-tuebingen.de (S.M.); michelle.johanna.daniela.dongo@med.uni-tuebingen.de (M.D.); claus.garbe@web.de (C.G.); 3Cluster of Excellence iFIT (EXC 2180), 72076 Tuebingen, Germany; 4Department of Dermatology, Charité-Universitätsmedizin Berlin, 10117 Berlin, Germany

**Keywords:** BRAF mutation, MAPK pathway, ERK inhibitor, drug resistance, melanoma therapy

## Abstract

Small molecule inhibitors targeting BRAF mutations at V600 and downstream MEK represent a significant advancement in treating BRAF-mutant melanoma. However, resistance mechanisms, often involving reactivation of the MAPK pathway via ERK1/2, limit their efficacy. The ERK1/2 inhibitor ravoxertinib (GDC0994) was tested on melanoma cell lines with and without resistance to BRAFi or BRAFi + MEKi. Short-term assays evaluated cell viability, apoptosis induction, and pathway modulation via Western Blot, while long-term effects were assessed using a colony formation assay. Resistant and parental melanoma cells responded to long-term ERKi treatment with reduced growth, independent of sensitivity to BRAF or MEK inhibitors. Adding ERKi to BRAFi/MEKi significantly enhanced apoptosis and growth inhibition, particularly in resistant lines. Although ravoxertinib alone showed limited antitumor activity, its combination with BRAFi/MEKi yielded substantial benefits, especially in chronic settings. These findings suggest that combinatorial regimens incorporating ERK inhibitors represent a promising therapeutic strategy for BRAF-mutant melanoma.

## 1. Introduction

The MAPK signaling pathway represents one of the major signal transduction cascades regulating various critical cellular processes including cell proliferation, differentiation and survival [1]. Therefore, it is not surprising that dysregulation of MAPK signaling is a common feature in many types of cancer and constitutes the basis for uncontrolled tumor growth [2,3]. Malignant melanoma is a prime example for a MAPK-driven tumor with approximately  50% of cutaneous melanomas harboring activating BRAF mutations and further cases exhibiting either activating mutations in RAS isoforms (20–30%) or loss-of-function mutations in the *NF1* tumor suppressor gene (10–15%), which serves as a negative regulator of MAPK pathway activity [4,5].

In line with this high prevalence of mutations activating MAPK signaling, targeting the MAPK pathway has proven to be an effective therapeutic strategy in melanoma. First preclinical and clinical evidence of this approach’s success came from the development of small molecule inhibitors selective for V600E/K-mutated BRAF (BRAFi), such as vemurafenib (PLX4032) and dabrafenib (GSK2118436), which revealed marked anti-tumor activity in BRAF-mutant melanomas [6,7]. The initially remarkable response rates, however, were limited by an inevitable and often rapidly developing resistance to this monotherapy [8]. While addition of MEK inhibitors (MEKi) such as trametinib (GSK1120212) or cobimetinib (GDC0973) could improve the duration of anti-tumor response and BRAFi/MEKi combinations are now standard of care following their FDA approval, therapeutic escape still occurs and frequently goes along with reactivation of the MAPK pathway [9,10,11,12]. Since ERK constitutes the most distal kinase and terminal effector of the MAPK signaling cascade, recent focus has been shifted to ERK inhibitors (ERKi) in order to overcome this resistance mechanism to inhibition of the upstream kinases BRAF and MEK (MAPKi) and to effectively target MAPK-dependent melanoma cells independent of an acquired BRAF/MEK inhibitor resistance [13,14,15,16].

Ravoxertinib (GDC0994) is an orally bioavailable and highly selective ERK1/2 inhibitor, which demonstrated a significant single-agent activity in different in vivo cancer models, including BRAF- and KRAS-mutant human colorectal and thyroid carcinoma xenografts in mice [17,18,19]. A first-in-human, phase I dose escalation study in patients with advanced solid tumors has further shown manageable safety profile and promising efficacy of GDC0994 in BRAF-mutant colorectal cancer [20]. Intriguingly, in contrast to other ERK inhibitors, such as VX-11e and SCH772984 GDC0994 has been reported to neither increase nor decrease ERK phosphorylation levels, which might be an indicator for alternative modes of action in terms of feedback signaling, and renders GDC0994 an interesting candidate to target MAPK-addicted cancer models including BRAFi/MEKi resistant malignant melanoma [13,20].

To explore the therapeutic potential of the ERKi GDC0994 in combination with BRAF and MEK inhibitors, this study evaluated its effects in preclinical melanoma models with and without acquired resistance to BRAF/MEK inhibition. In summary, our data suggests that additional ERK inhibition by GDC0994 may help to enhance MAPK pathway suppression and could offer a potential strategy to overcome or delay resistance in melanoma cells.

## 2. Results

### 2.1. ERKi Is Able to Block Downstream MAPK Signalling in BRAF-Mutated Cell Lines Irrespective of BRAFi Resistance

To investigate whether the ERK1/2 inhibitor GDC0994 is able to block the MAPK pathway by suppression of ERK downstream signaling, BRAF mutated melanoma cell lines were treated with the ERK inhibitor (ERKi) and the inhibitor vemurafenib (PLX4032), which is specific for mutated BRAF (BRAFi). In BRAF-mutant melanoma cell lines, which are sensitive to the clinically approved BRAFi PLX4032 (S), ERKi robustly attenuated MAPK pathway activity to a similar extent as PLX4032, as reflected by reduced phosphorylation of RSK, a direct target of ERK1/2 (Figure 1A). As expected, the GDC0994 did not majorly impact phosphorylation of ERK itself and of its upstream kinase MEK in contrast to treatment with the BRAF inhibitor, which blocks the phosphorylation of MEK in both sensitive cell lines tested and also blocks the phosphorylation of ERK in SKMel19 S. In A375 S the phosphorylation of ERK is also reduced but to a lesser extent.

To assess if GDC0994 can also effectively block MAPK signaling in cells resistant to BRAF inhibition, melanoma cells with acquired BRAFi resistance (R), that were generated through chronic vemurafenib (PLX4032) exposure, were treated with ERKi or BRAFi. In line with a common hyperactivation of MAPK signaling in acquired resistance to BRAF inhibitors in a clinical setting [8,11], increased phosphorylation levels of ERK and RSK have been observed previously in the resistant cell lines compared to their BRAFi sensitive counterparts [21]. Strikingly, ERKi retained its inhibitory effect in the BRAFi-resistant setting, effectively suppressing ERK-downstream signaling, reflected by decreased RSK phosphorylation, despite the acquired MAPKi resistance, as evidenced by the lack of PLX4032-mediated inhibition of MEK, ERK, and RSK phosphorylation in these cells (Figure 1B). These data underscore the potential of ERKi to overcome resistance mechanisms that emerge under chronic BRAFi treatment.

### 2.2. BRAFi + ERKi Combinations Reduce Cell Growth and Induce Apoptosis, Especially in MAPKi Resistant Cell Lines

To test the effects of the ERKi as a monotherapy or in combination with the BRAFi on the viability of different BRAF-mutated melanoma cell lines (451LU, Mel1617, A375), dose-response curves were performed. PLX4032 and GDC0994 were used at a range from 0.039 to 10 µM. As expected, monotherapy with the BRAF inhibitor PLX4032 (black line) showed strong effects on the parental BRAF-mutated cells still sensitive to BRAFi (Figure 2A, upper panel). The ERKi also led to a decrease in viability in all three tested cell lines (brown line), however, with reduced efficacy compared to PLX4032. Similarly, the combinational treatment was not able to achieve an additive or synergistic effect (orange line). In the lower panel of Figure 2A, the resistant counterparts of the cell lines were treated with the same inhibitors and their combination. As expected, because these cell lines are resistant to the BRAFi due to chronic treatment, they showed only a decrease in viability after treatment with very high concentrations of PLX4032. While ERKi monotherapy again could not strongly reduce melanoma cell viability, the combinational treatment with GDC0994 and PLX4032 was able to achieve a synergistic enhancement of the inhibitory effects in the melanoma cell lines with acquired BRAF inhibitor resistance. Combination index analysis (Figure 2B) showed a synergistic effect (CI < 1) for most of the combinations of PLX4032 and GDC0994 in 451LU, Mel1617 and A375 cell lines. Of note, it was seen in both the parental S cell line, as well as in the corresponding resistant cell line.

Intriguingly, additional cell cycle analysis revealed comparable efficiencies in apoptosis induction by PLX4032 and GDC0994 monotherapies in the BRAFi sensitive cell lines, reflected by an enhanced fraction of cells in sub-G1 (Figure 2C, upper panel, red section of the bar). The effect of PLX4032 on apoptosis induction in the sensitive cell lines could be slightly enhanced by addition of the ERKi GDC0994. After combinational treatment, 40% (with 1 µM PLX4032 and 1 µM GDC0994) and 56% (with 5 µM PLX4032 and 5 µM GDC0994) of 451LU S cells were found in the subG1 fraction (Figure 2C, upper panel). For Mel1617 S, the combinational treatments led to 37.0% (with 1 µM PLX4032 and 1 µM GDC0994) and 42% (with 5 µM PLX4032 and 5 µM GDC0994) of cells in the subG1 fraction (Figure 2C, upper panel). The last cell line tested, A375 S, showed apoptotic cell fractions of 45.0% (with 1 µM PLX4032 and 1 µM GDC0994) and 46% (with 5 µM PLX4032 and 5 µM GDC0994) after combination treatment (Figure 2C, upper panel). Most importantly, the ERK inhibitor was still capable of inducing apoptosis in the BRAF inhibitor resistant cell lines, albeit to a lesser extent compared to the respective sensitive counterparts and predominantly visible at the higher tested inhibitor concentration (Figure 2C, lower panel). Indeed, combinational treatments with the BRAF and ERK inhibitors, especially at 5 µM inhibitor concentrations, could robustly induce apoptosis in the BRAFi resistant cells. While in 451LU R around 32% of cells could be detected in sub-G1, apoptotic cell fractions induced by combinational treatment in Mel1617 R (27.0% with 1 µM PLX4032 and 1 µM GDC0994 and 54% with 5 µM PLX4032 and 5 µM GDC0994) and A375 R (63.0% with 1 µM PLX4032 and 1 µM GDC0994 and 56% with 5 µM PLX4032 and 5 µM GDC0994) were comparable to or even exceeded those in the sensitive counterparts (Figure 2C). A potential usefulness of the BRAFi + MEKi combination treatment in terms of apoptosis induction in both BRAF inhibitor sensitive and resistant cell lines was further supported by additional flow cytometric analysis of apoptotic cells detecting phosphatidylserine residue translocation to the cell surface (Appendix A).

### 2.3. BRAFi + ERKi Inhibits Long Term Cell Growth

In a 10-day long-term colony formation assay, the BRAF-mutant melanoma cell lines 451LU, Mel1617, A375 and SKMel19—either parental BRAFi-sensitive (S) or with BRAFi-resistance (R) through chronic exposure to PLX4032—were treated with the ERK inhibitor GDC0994 either as monotherapy or in addition to ongoing BRAFi treatment. The ERK inhibitor, particularly when administered in the combinatorial approach, resulted in a significant suppression of colony formation in both sensitive and resistant cells, indicating that (additional) ERKi can effectively impair long-term proliferative capacity even in the context of acquired BRAFi resistance (Figure 3A,B). Notably, chronic treatment with GDC0994 in combination with PLX4032 led to nearly no visible colonies across all cell lines tested, indicating an enhanced efficacy of the ERK inhibitor with prolonged administration.

### 2.4. Additional ERKi Induces Apoptosis and Inhibits Long Term Cell Grwoth in Double-Resistant Cell Lines

The treatment with a combination of BRAFi and MEKi represents the standard therapy for BRAF-mutated patients in the clinic nowadays next to immunotherapy. Therefore, the ERKi inhibitor was also tested in cell lines which were rendered resistant against BRAFi + MEKi (double-resistant). Consistent with our findings in melanoma cell lines with acquired resistance to BRAFi monotherapy, the addition of the ERK inhibitor GDC0994 helped to improve the response of Mel1617 double-resistant cells (Mel1617 RR) to the standard BRAFi + MEKi combination treatment (PLX4032 + GDC0973) as demonstrated by dose-response cell viability assays (Figure 4A, top panel), whereas treatment with ERKi as a single agent exerted only minimal effects on their cell viability. Accordingly, combination index analysis (Figure 4A, bottom panel) showed a synergistic effect (CI < 1) for several combinations of PLX4032+GDC0973 and GDC0994. Additional cell cycle analysis in Mel1617 RR and A375 RR further revealed that the triple combination treatment (PLX4032 + GDC0973 + GDC0994) markedly enhanced apoptosis induction (Figure 4B) leading to sub-G1 fractions up to 23% (Mel1617 RR), while neither treatment with the drugs as single agents nor the BRAF and MEK inhibitor combination alone had a visible influence on apoptotic cell fractions. Similarly, monotherapies with the BRAF, MEK or ERK inhibitor as well as the BRAF and MEK inhibitor combination were not effective in suppressing the growth of the double-resistant melanoma cell lines in a long-term colony formation assay and for A375 RR partly even promoted cell growth (Figure 4C). However, administration of a triple combination treatment adding the ERK inhibitor to the standard BRAF and MEK inhibitor combination was capable of strongly reducing long-term cell growth of both double-resistant melanoma cell lines following 10 days of treatment, as evidenced by a decreased number of colonies in colony formation assays after triple treatment (Figure 4C). This enhanced efficacy of the triple combination was further supported by reduced MAPK pathway activity as reflected by decreased phosphorylation of both ERK and its downstream target RSK, while neither the monotherapies nor the BRAFi + MEKi combination alone produced comparable effects (Appendix A).

Taken together, our data support the notion that simultaneous targeting of multiple nodes within the MAPK signaling cascade (Figure 5) may enhance therapeutic efficacy and contribute to overcoming acquired resistance mechanisms.

## 3. Discussion

ERK1/2 inhibition represents a promising strategy for targeting tumors with aberrant MAPK signaling, particularly those harboring BRAF or RAS mutations. GDC-0994, a selective, orally bioavailable ERK1/2 inhibitor, was developed to address limitations of upstream inhibition with RAF or MEK inhibitors, such as rapid feedback reactivation of the pathway and development of acquired resistance [17]. Indeed, various small molecule inhibitors of ERK1/2 such as SCH772984, VX-11e and BVD-523 (Ulixertinib) have been developed and used to effectively target MAPK-dependent melanoma cells in preclinical model systems [13,14,15]. Additional studies support the efficacy of ERK inhibitors in MAPKi resistant models. Hatzivassiliou et al. [22] demonstrated that ERK inhibition in cells with acquired resistance to MEK inhibitors could effectively block MAPK signaling in breast and colorectal adenocarcinoma. Moreover, it could be shown that ERK inhibition with VX-11e could overcome resistance to RAF/EGFR or RAF/MEK inhibitor combination therapies in BRAF-mutant colorectal cancer by sustaining suppression of MAPK pathway activity [23]. Krepler et al. [24] further investigated combinations with VX-11e in patient-derived xenograft (PDX) melanoma models following relapse under BRAF inhibitor therapy, highlighting the potential clinical relevance of this approach.

As an ATP-competitive type I inhibitor, GDC-0994 binds to the active conformation of ERK1/2 and suppresses downstream signaling responsible for transcription and proliferation as well as invasion of cancer cells [25,26,27,28]. Preclinical studies demonstrated that GDC-0994 effectively inhibits the growth of RAS- and BRAF-mutant cancer cells, including next to colorectal cancer also melanoma and thyroid cancer, without significantly altering ERK phosphorylation (P-ERK) levels [18,19,29]. This distinguishes GDC-0994 from other ERK inhibitors, such as VX-11e, Ulixertinib or SCH772984, which increase or inhibit P-ERK levels, respectively [13,15,20]. Indeed, GDC-0994, VX-11e and SCH772984 each belong to a distinct scaffold class with different underlying modes of action [30]. SCH772984, for example, modulates P-ERK via dual binding to active and inactive ERK conformations [13]. Interestingly, the lack of P-ERK modulation by GDC-0994 may reduce compensatory feedback activation, but it also presents challenges in using pERK as a biomarker of efficacy.

Further supporting the biological and clinical relevance of selecting ERK1/2 inhibitors from distinct scaffold classes, the mechanisms underlying acquired resistance to GDC-0994 appear to differ from those associated with other ERK inhibitors [30]. Unlike SCH772984or VX-11e, which often face resistance due to ERK1/2 mutations, GDC0994-resistant cells did not harbor ERK point mutations, but instead showed ERK2 and MITF amplifications. Notably, both ERK2 amplifications and mutated ERK1/2 retained sensitivity to MEK inhibition, indicating potential for sequential or combination treatment with upstream MAPK inhibitors. Furthermore, scaffold-specific resistance implies that switching to ERK inhibitors with different binding modes—such as GDC0994—may help overcome acquired resistance to previous ERK inhibition and could provide a valuable strategy to overcome or delay resistance in cancer therapy. In clinical evaluation, GDC-0994 showed partial MAPK pathway suppression (19–51%) in patients with BRAF- and KRAS-mutant tumors, but responses were modest with only one third of patients having a best overall response of stable disease and approximately 5% with confirmed partial response [20]. The limited clinical activity may be attributable to suboptimal pathway inhibition, as prior studies suggested that full suppression of ERK activity is necessary for tumor regression [31]. Adaptive resistance mechanisms, including feedback activation via EGFR or ERK2/MITF amplifications, may also undermine monotherapy efficacy [20,30]. Therefore, to achieve more sustained and comprehensive inhibition of MAPK signaling, combinatorial strategies simultaneously targeting multiple MAPK pathway nodes, such as ERK, MEK and BRAF may be warranted (Figure 5).

In support of this strategy, combined BRAF, EGFR, and MEK inhibition has demonstrated superior MAPK pathway suppression and improved clinical outcomes compared to the respective monotherapies in patients with BRAF-mutated colorectal carcinoma [32]. In malignant melanoma, the concurrent use of BRAF and MEK inhibitors to simultaneously target different nodes in the MAPK pathway has already proven to significantly enhance both the depth and durability of treatment responses [9,10,12]. Additional preclinical evidence suggests that ERK inhibitors may further augment the therapeutic efficacy of MAPK-targeted regimens [13,15,33]. However, clinical combination of the ERK inhibitor GDC0994 with the MEK inhibitor cobimetinib in patients with advanced solid tumors led to intolerable overlapping toxicities, limiting its feasibility [34]. Despite this, vertical pathway inhibition using different combinations (e.g., BRAF plus ERK) remains a promising avenue to achieve deeper, sustained suppression. This should be investigated in greater detail in future studies, both in vitro and in vivo, with the goal to fully understand the underlying functional mechanisms and potential escape pathways of the cancer cells along with identifying the most effective combination strategies for clinical application.

Altogether, GDC-0994 exemplifies the potential and limitations of targeting ERK1/2. While preclinical data support its role in suppressing MAPK signaling and delaying resistance development, its clinical activity as monotherapy has been modest, likely due to incomplete target engagement and adaptive signaling. Scaffold-specific resistance patterns further underscore the need for rational sequencing and combination of ERK inhibitors. Future strategies may require not only improved ERK inhibitors but also optimized schedules and combinations tailored to tumor genotype and resistance profiles.

## 4. Materials and Methods

### 4.1. Culture of Human Cells

The BRAF^V600E^-mutated melanoma cell lines A375 and SKMel19 were purchased from ATCC (LGC, Wesel, Germany), while Mel1617 and 451LU were kindly provided by M. Herlyn (The Wistar Institute, Philadelphia, PA, USA) [35]. The cells were cultured at 37 °C with 5% CO_2_ and 95% humidity using RPMI 1640 medium supplemented with 10% fetal calf serum (FCS) and 1% penicillin-streptomycin. Potential mycoplasma infection of the cells was regularly tested with the Venor GeM Classic Mycoplasma Detection Kit (Minerva Biolabs, Berlin, Germany) and cells were used for experiments within two months of thawing from the frozen stock. All experiments were performed in accordance with the Declaration of Helsinki Principles.

The generation of melanoma cells with acquired resistance to the BRAF inhibitor vemurafenib (R) or dual resistance to BRAF and MEK inhibitors (vemurafenib and cobimetinib, RR) was conducted as described previously [36]. To minimize direct inhibitor effects, resistant cells were maintained in inhibitor-free culture medium for 24 h prior to experiments.

### 4.2. Signaling Pathway Inhibitors and Treatments

Vemurafenib (PLX4032), cobimetinib (GDC0973) and ravoxertinib (GDC0994) were purchased from Selleck Chemicals (VWR, Darmstadt, Germany) and dissolved in dimethylsulfoxide (DMSO).

### 4.3. Viability Assay

Melanoma cell viability was assessed using the 4-methylumbelliferyl heptanoate (MUH) assay. Briefly, 2.5 × 10^3^ cells were seeded into 96-well plates. After 24 h, cells were treated in hexaplicates for 72 h with increasing concentrations of vemurafenib (up to 20 µM) or ravoxertinib (up to 20 µM) either as single agents or in combination. Cells were washed with phosphate buffered saline (PBS) and incubated for 1 h at 37 °C with 100 μg/mL 4-methylumbelliferyl heptanoate diluted in PBS prior to analysis. In viable cells, intracellular esterases and lipases hydrolyze 4-methylumbelliferyl heptanoate, generating the highly fluorescent 4-methylumbelliferone. Fluorescence (λ_ex_ 355 nm, λ_em_ 460 nm) was measured using a Tristar fluorescence microplate reader (Berthold Technologies, Bad Wildbad, Germany). The fluorescence intensity correlates with the number of viable cells. The relative viable cell number following treatment was calculated by normalization to the solvent controls.

### 4.4. Cell Cycle Analysis

For cell cycle analysis, 2.5 × 10^5^ cells were seeded into 6-well plates. After incubation for 16 h, the cells were treated with the indicated concentrations and combinations of vemurafenib, cobimetinib and ravoxertinib for 3 days. Treatment was performed in triplicates, with DMSO (0.02% *v*/*v*) serving as a solvent control. To assess the cell cycle distribution, floating and adherent cells were harvested, permeabilized overnight with 70% ice-cold ethanol, washed twice with PBS, and resuspended in PBS supplemented with 50 μg/mL propidium iodide (Sigma-Aldrich, Darmstadt, Germany) and 100 μg/mL RNAse A (AppliChem, Darmstadt, Germany). After a 30 min-incubation in the dark, the cell cycle distribution was recorded using a BD^TM^ LSR II flow cytometer (BD Biosciences, Heidelberg, Germany) and FACSDivaTM software (Version 6.1.3) (BD Biosciences, Heidelberg, Germany).

### 4.5. Flow Cytometric Apoptosis Assay

Following treatment, cells were harvested, washed twice with phosphate-buffered saline (PBS), and stained with Apotracker Green (BioLegend, San Diego, CA, USA) and Zombie NIR Viability Dye (BioLegend, San Diego, CA, USA) according to the manufacturer’s instructions to detect apoptotic cells for 30 min. After incubation, cells were then washed and resuspended in PBS for flow cytometric analysis. Samples were acquired using a Cytek Aurora flow cytometer (Cytek Biosciences, Fremont, CA, USA). A minimum of 100,000 events were collected per sample. Compensation was performed using single-stained controls (using staurosporin treatment as an apoptosis inducer). Data were analyzed using FlowJo v10. Cells were gated to exclude debris and doublets. Based on fluorescence signals, cells were categorized into live (Apotracker^−^/Zombie NIR^−^; lower left panel), early apoptotic (Apotracker^+^/Zombie NIR^−^; upper left panel), and late apoptotic/necrotic (Apotracker^+^/Zombie NIR^+^; upper right panel) populations.

### 4.6. Clonogenic Assays

Cells were seeded into a 12-well plate at low density (200 cells per cavity) and treated with signaling pathway inhibitors as indicated. Culture medium containing the respective inhibitors was exchanged twice a week. After 10 days, cells were fixed with 4% paraformaldehyde and colonies visualized by staining with a 0.1% Coomassie Brilliant Blue solution (Bio-Rad, Feldkirchen, Germany) containing 30% methanol and 10% acetic acid.

### 4.7. Western Blot

After treatment with the indicated concentrations of vemurafenib, cobimetinib and ravoxertinib for 24 h, melanoma cells were harvested, washed twice with PBS and incubated on ice for 30 min with lysis buffer (10 mmol/L Tris (pH 7.5), 0.5% Triton X-100, 5 mmol/L EDTA, 0.1 mmol/L phenylmethylsulfonyl fluoride, 10 mmol/L pepstatin A, 10 mmol/L leupeptin, 25 mmol/L aprotinin, 20 mmol/L sodium fluoride, 1 mmol/L pyrophosphate, 1 mmol/L orthovanadate). Insoluble fractions were cleared by centrifugation at 13,000× *g* for 30 min and protein content of the lysates was measured using Bradford reagent (Bio-Rad, Feldkirchen, Germany). 30 µg of protein per sample was supplemented with Lämmli buffer, subjected to SDS-PAGE and transferred onto polyvinylidene difluoride (PVDF) membranes. Antibodies against MEK1/2 (#4694), p-MEK1/2 (S217/S221) (#9154), ERK1/2 (#9102), p-ERK (T202/Y204) (#4370), RSK1-3 (#9355), p-RSK (T359/S363) (#9344) and GAPDH (#2118) (Cell Signaling Technology, Leiden, The Netherlands) were applied and detected using anti-rabbit (#7074) or anti-mouse IgG (#7076) horseradish peroxidase (HRP)-conjugated secondary antibodies (Cell Signaling Technology, Leiden, The Netherlands). Chemiluminescent signals produced by PierceTM ECL Western Blotting Substrate or the SuperSignal^TM^ West Dura Extended Duration Substrate (both Thermo Fisher Scientific, Waltham, MA, USA) were captured on X-ray films (Eastman Kodak, Rochester, New York, NY, USA).

### 4.8. Statistics and Combination Index (CI) Analysis

To assess the potential synergy of inhibitor combinations, we utilized CompuSyn software Version 1.0 from ComboSyn. Inc. (Paramus, NJ, USA) to calculate combination indices (CIs). CI values of 1 denote additive effects, whereas values below 1 denote synergistic effects and values above 1 To quantitatively assess drug interactions, we performed combination index (CI) analyses using the CompuSyn software (ComboSyn Inc., Paramus, NJ, USA), which is based on the median-effect principle and the Chou-Talalay method [34]. Briefly, cells were treated with serial dilutions of each inhibitor alone and in fixed-ratio combinations at 1:1 molar ratio of PLX4032 and GDC0994, followed by cell viability measurement after 72 h using the MUH viability assay. For double-resistant cells the PLX4032-GDC0973 combination treatment has been used instead of the PLX4032 single treatment. Viability data were normalized to controls, and the dose–effect relationship for each (single) agent and combination was analyzed using CompuSyn to calculate the CI values at different effect levels (fraction affected, Fa). CI values were interpreted as follows: CI = 1 indicates an additive effect, CI < 1 denotes synergism, and CI > 1 indicates antagonism [37].

## 5. Conclusions

The combination of ERKi with BRAFi/MEKi represents a potent strategy to target MAPK pathway reactivation in melanoma. Future studies should explore the clinical translation of this approach to improve outcomes for patients with BRAF-mutant melanoma.

Our findings demonstrate that combining ERK inhibitors with BRAF and MEK inhibitors can enhance suppression of MAPK signaling, particularly in melanoma cells with acquired resistance to BRAFi/MEKi. This combinatorial approach holds promise for overcoming key therapeutic resistance mechanisms in BRAF-mutant melanoma. Future studies are warranted to further evaluate the therapeutic potential and clinical feasibility of this strategy, with the goal of improving treatment durability and patient outcomes, while at the same time limiting toxicities.

## Figures and Tables

**Figure 1 ijms-26-09794-f001:**
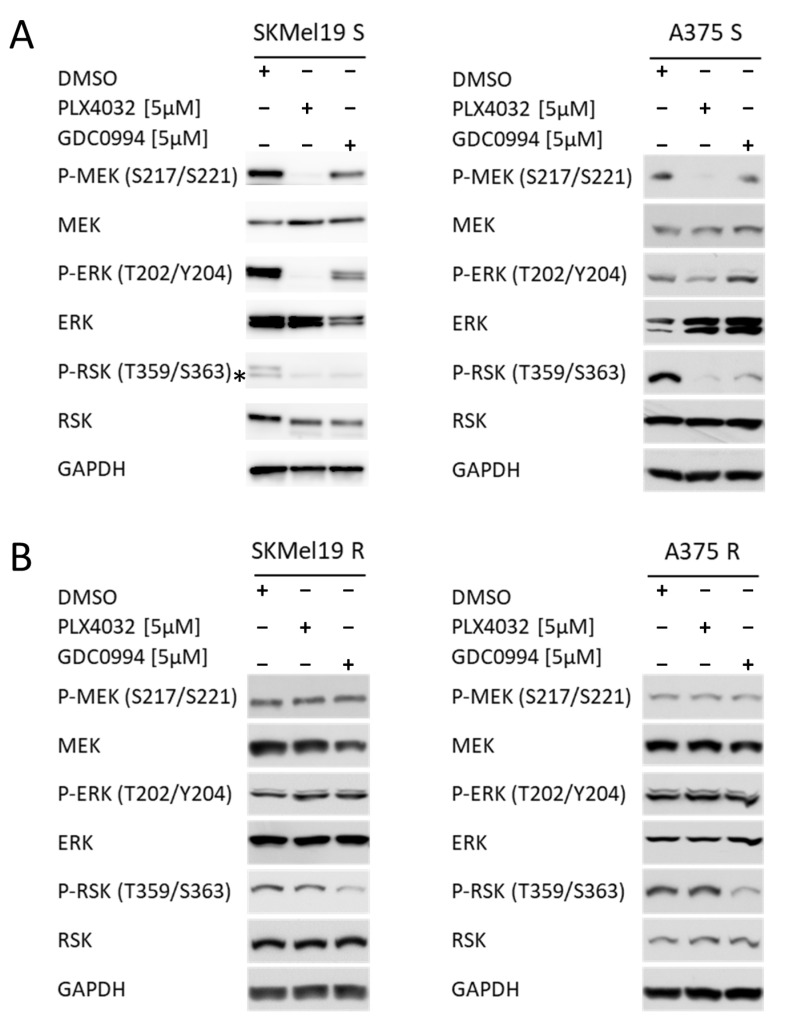
Phosphorylation status of MEK, ERK and RSK. (**A**) Western blot analysis of p-MEK/MEK, p-ERK/ERK and p-RSK/RSK in SKMel19 S and A375 S cells. GAPDH was used as a reference protein, and the samples were loaded on a 10% SDS polyacrylamide gel. Cells were treated for 24 h with 5 µM PLX4032 or 5 µM GDC0994. DMSO treatment was used as a control sample. (**B**) Western blot analysis of p-MEK/MEK, p-ERK/ERK and p-RSK/RSK in SKMel19 R and A375 R cells. GAPDH was used as a reference protein, and the samples were loaded on a 10% SDS polyacrylamide gel. Cells were treated for 24 h with 5 µM PLX4032 or 5 µM GDC0994. DMSO treatment was used as a control sample. (* unspecifc off target band).

**Figure 2 ijms-26-09794-f002:**
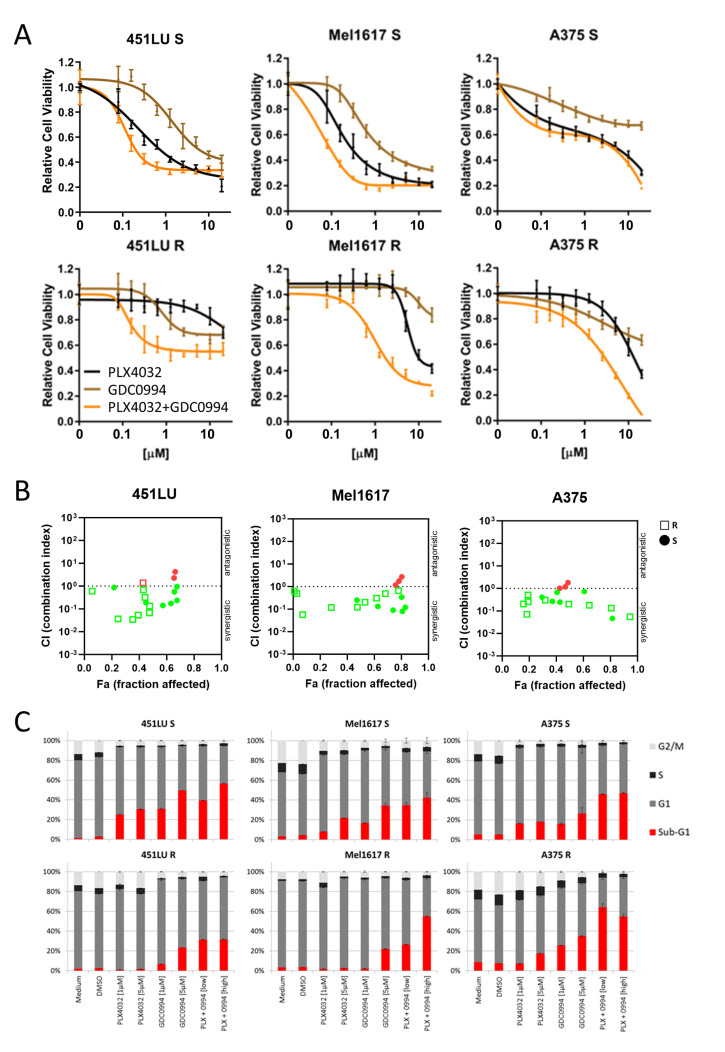
Cell viability and cell cycle. Effects of the combination of ERKi/BRAFi on the viability (**A**,**B**) and cell cycle (**C**) of melanoma cells. (**A**) 451LU S, 451LU R, Mel1617 S, Mel1617 R, A375 S and A375 R cells were treated with BRAFi (PLX4032) (up to 20 µM), ERKi (GDC0994) (up to 20 µM) and the combination for 72 h. Measured is the viability of treated cells compared to the DMSO-treated controls. Shown are the mean values with standard deviations (SDs) of three independent experiments, each measured in quadruplicate and normalized to the untreated control. (**B**) Shown is the synergism analysis of 451LU (S and R), Mel1617 (S and R) and A375 (S and R) treated with BRAFi (PLX4032) (up to 20 µM), ERKi (GDC0994) (up to 20 µM) and the combination for 72 h. (red = antagonistic, green = synergistic). (**C**) 451LU S, 451LU R, Mel1617 S, Mel1617 R, A375 S and A375 R cells were treated with PLX4032 (1 or 5 µM), GDC0994 (1 and 5 µM) and the combination (1 µM + 1 µM or 5 µM + 5 µM) for 72 h. The cell cycle after PI staining was measured by flow cytometry. Shown are the mean values of each cell cycle fraction with SDs of three independent experiments, each measured in triplicate.

**Figure 3 ijms-26-09794-f003:**
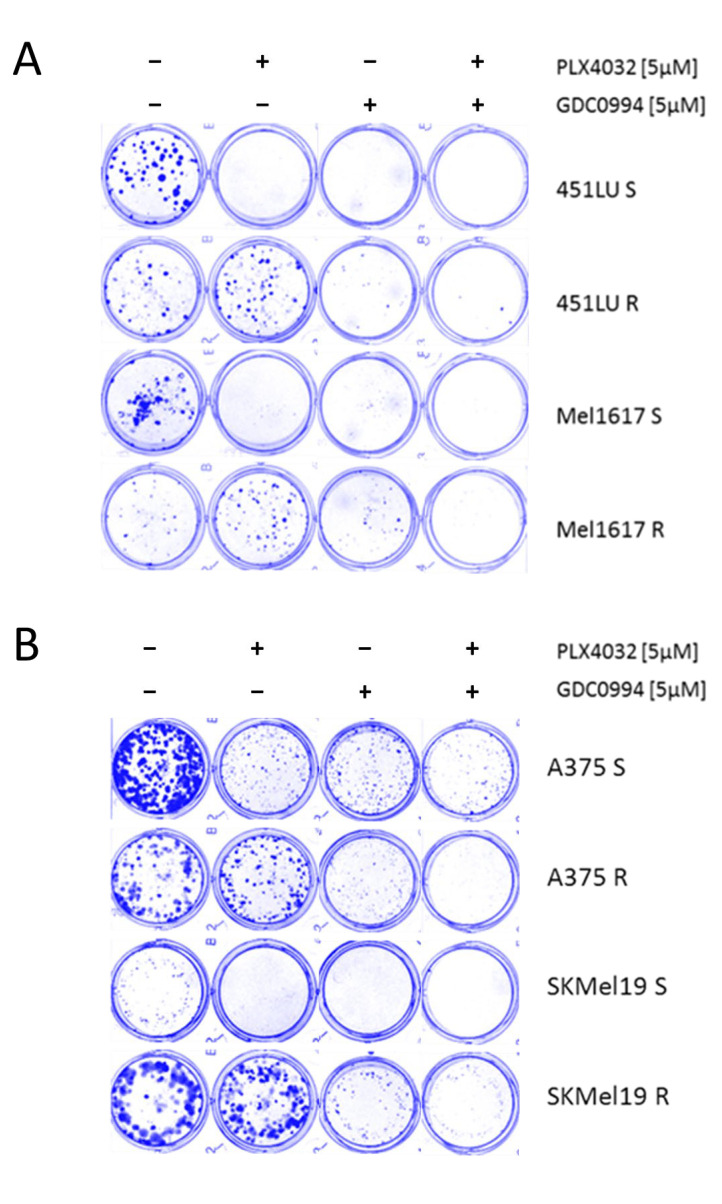
Colony formation assay. (**A**) The cell lines 451LU S, 451LU R, Mel1617 S and Mel1617 R and (**B**) the cell lines A375 S, A735 R, SKMel19 S and SKMel19 R were seeded and then treated for 10 consecutive days with PLX4032 (5 µM), GDC0994 (5 µM) or the combination. The colonies were fixed with 4% paraformaldehyde visualized by staining with a 0.1% Coomassie Brilliant Blue solution (Bio-Rad) containing 30% methanol and 10% acetic acid.

**Figure 4 ijms-26-09794-f004:**
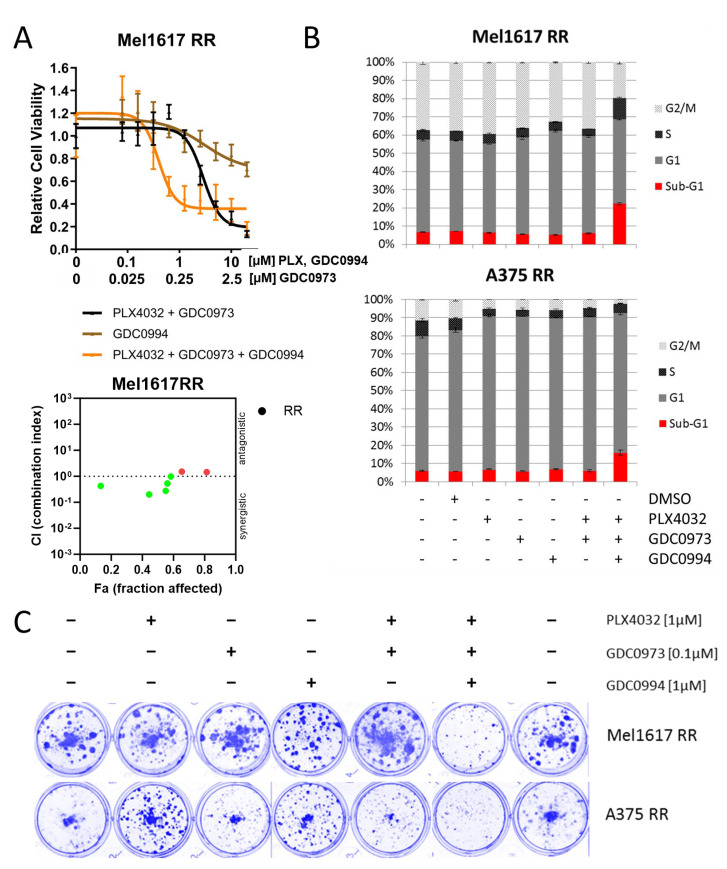
Effect on double-resistant cell lines. (**A**) Mel1617 RR cells were treated with PLX4032+GDC0973 (up to 20 µM PLX4032 and 5 µM GDC0973), GDC0994 (up to 20 µM) and the combination for 72 h. Measured is the viability of treated cells compared to the DMSO-treated controls. Shown are the mean values with standard deviations (SDs) normalized to the untreated control. The bottom panel shows the corresponding synergism analysis. (red = antagonistic, green = synergistic). (**B**) A375 RR and Mel1617 RR cells were treated with PLX4032 (1 µM), GDC0994 (1 µM), GDC0973 (0.1 µM) and the combination thereof for 72 h. The cell cycle after PI staining was measured by flow cytometry. Shown are the mean values of each cell cycle fraction with SDs of three independent experiments, each measured in triplicate. (**C**) The cell lines A375 RR and Mel1617 RR were seeded and then treated for 10 consecutive days with PLX4032 (1 µM), GDC0994 (1 µM), GDC0973 (0.1 µM) or their combination. The colonies were fixed with 4% paraformaldehyde and visualized by staining with a 0.1% Coomassie Brilliant Blue solution (Bio-Rad) containing 30% methanol and 10% acetic acid.

**Figure 5 ijms-26-09794-f005:**
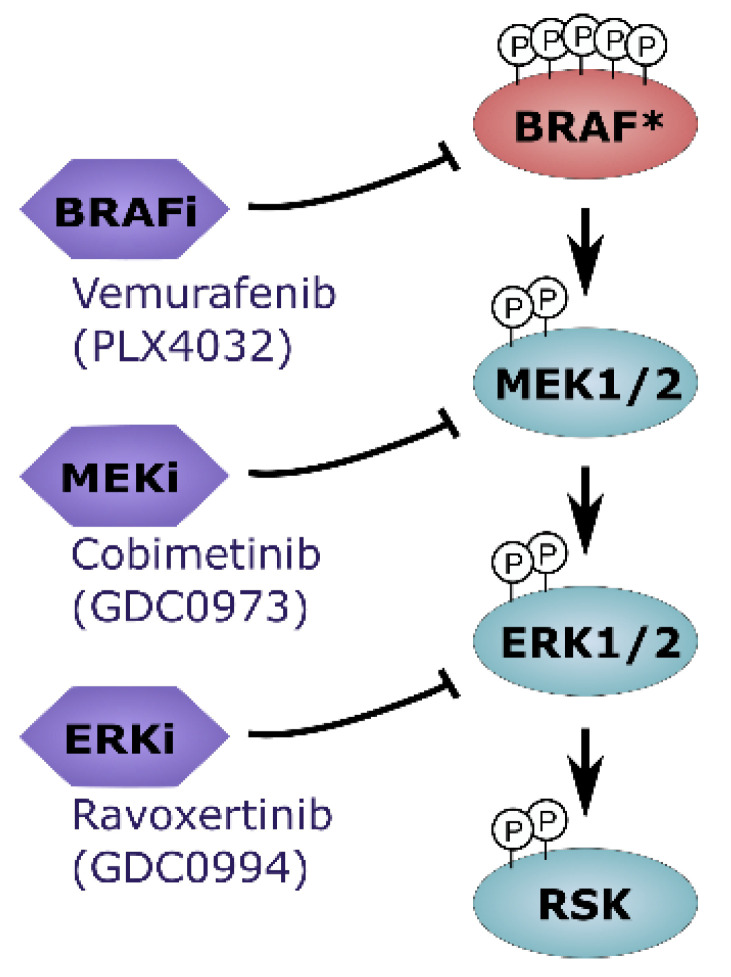
Overview on in-depth MAKP pathway inhibition. Combinatorial treatment regimens that include ERK1/2 inhibitors may represent an attractive novel therapeutic strategy for BRAF-mutated melanoma cells (* mutated BRAF).

## Data Availability

Data is contained within the article or Appendix A.

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
