# Peer review of "Combinatorial ERK Inhibition Enhances MAPK Pathway Suppression in BRAF-Mutant Melanoma"

_ijms, 2025, doi:10.3390/ijms26199794_

Round 1

Reviewer 1 Report

Comments and Suggestions for Authors

The authors reported that the ERK1/2 inhibitor ravoxertinib (GDC0994) reduced cell growth in resistant and parental melanoma cells. GDC0994 and PLX4032 combinations reduce cell growth and induce apoptosis, especially in MAPKi resistant cell lines. Lack of animal experiment, reuse of data, and arbitrary conclusions hinder the quality and credibility of the manuscript.

Specific issues include:

1) Is there a difference in the phosphorylation level of BRAF downstream targets (ERK and RSK) between BRAFi sensitive and resistant cell lines?

2) Fig 2A, it seems that cell viability curves of the GDC group in 451LU S and 451LU R were same.

3) In addition to cell line-based experiments, in vivo experiments are also needed to confirm the effectiveness of the combination of the two drugs (PLX4032 and GDC0994).

4) Regarding the effect of drugs on cell apoptosis, apoptosis flow cytometry (Annexin V/PI double staining) and detection of apoptosis marker molecule are also needed.

5) Fig 4B showed that PLX4032 or GDC0994 alone increased long-term cell growth of the double resistant melanoma cell lines following 10 days of treatment, which inconsistent with the description.

6) More recent published literature about GDC0994 or other ERKi should be citated and discussed [PMID: 39389321; 38728872].

Author Response

1) Is there a difference in the phosphorylation level of BRAF downstream targets (ERK and RSK) between BRAFi sensitive and resistant cell lines?

We thank the reviewer for pointing out this important aspect. In line with the common hyperactivation of the MAPK signalling pathway in the framework of acquired resistance to BRAF inhibitors (PMID: 21383288, PMID: 24265155), we could observe both increased phosphorylation levels of ERK and RSK in our resistant cell lines compared to their BRAFi sensitive counterparts. This we have already published in previous work: Kosnopfel et al., 2017, Oncotarget (PMID: 28415756)

2) Fig 2A, it seems that cell viability curves of the GDC group in 451LU S and 451LU R were same.

We apologize for this mistake: We changed this graph accordingly.

3) In addition to cell line-based experiments, in vivo experiments are also needed to confirm the effectiveness of the combination of the two drugs (PLX4032 and GDC0994).

We agree with the reviewer that in vivo validation would further strengthen our findings. However, due to the time and resource constraints of the current study, we have focused on in vitro analyses using melanoma cell lines to establish the efficacy and synergistic interaction of PLX4032 and GDC0994. We fully acknowledge the importance of in vivo studies and have revised the Discussion section to highlight this limitation and our ongoing efforts.

4) Regarding the effect of drugs on cell apoptosis, apoptosis flow cytometry (Annexin V/PI double staining) and detection of apoptosis marker molecule are also needed.

In accordance with the suggestion of the reviewer, we included flow cytometric stainings of apoptotic cells using Apotracker Green (detecting the translocation of phosphatidylserine residues to the cell surface) and Zombie NIR to corroborate a potential apoptosis induction by the drugs, as indicated by the elevated sub-G1 fractions in the previously conducted cell cycle analyses. The new Supplementary Figure S1 shows the results for both parental sensitive melanoma cell lines and their respective counterparts with acquired BRAF inhibitor resistance after treatment with PLX4032, GDC0994 or the combination thereof.

5) Fig 4B showed that PLX4032 or GDC0994 alone increased long-term cell growth of the double resistant melanoma cell lines following 10 days of treatment, which inconsistent with the description.

We thank the reviewer for this valuable remark and described this interesting finding in more detail in our results section, which now reads as follows:

“Similarly, monotherapies with the BRAF, MEK or ERK inhibitor as well as the BRAF and MEK inhibitor combination were not effective in suppressing the growth of the double resistant melanoma cell lines in a long-term colony formation assay and for A375 RR partly even promoted cell growth (Figure 4C). However, administration of a triple combination treatment adding the ERK inhibitor to the standard BRAF and MEK inhibitor combination was capable of strongly reducing long-term cell growth of both double resistant melanoma cell lines following 10 days of treatment, as evidenced by a decreased number of colonies in colony formation assays after triple treatment (Figure 4C).”

6) More recent published literature about GDC0994 or other ERKi should be citated and discussed [PMID: 39389321; 38728872].

We fully agree and included the respective publications in our introduction and discussion sections.

Reviewer 2 Report

Comments and Suggestions for Authors

I read the article  "Combinatorial ERK Inhibition Enhances MAPK Pathway Suppression in BRAF Mutant Melanoma" with great interest.  Authors have tried to explore the effect of ERK inhibition on the most common drug resistance type of melanoma. 

Though experimental data primarily support the conclusion, the author still needs to do many other experiments to confirm the findings. Also, there are already papers that have shown a similar outcome using GDC0994. However, it would be interesting to explore the underlying details mechanism. Confirmatory experiments using knockdown and overexpression for rescue, in vivo resistance development, and targeting melanoma using GDC0994 would be helpful.

Thank you

Author Response

I read the article "Combinatorial ERK Inhibition Enhances MAPK Pathway Suppression in BRAF Mutant Melanoma" with great interest.  Authors have tried to explore the effect of ERK inhibition on the most common drug resistance type of melanoma.

Though experimental data primarily support the conclusion, the author still needs to do many other experiments to confirm the findings. Also, there are already papers that have shown a similar outcome using GDC0994. However, it would be interesting to explore the underlying details mechanism. Confirmatory experiments using knockdown and overexpression for rescue, in vivo resistance development, and targeting melanoma using GDC0994 would be helpful.

We thank the reviewer for her/his interest in our research and her/his invaluable comments. We completely agree that further experiments are warranted to extend our understanding of the underlying functional mechanisms and to assess the safety and usefulness of our proposed inhibitor combinations in vivo and in the clinical setting (we included this in the discussion). However, this is beyond the scope of the presented manuscript and will be addressed in future studies.

Although similar studies have been conducted assessing the usefulness of GDC0994 in BRAF mutated cancers including malignant melanoma, to the best of our knowledge, no study has investigated the potential benefit of adding the ERK inhibitor GDC0994 to MAPK pathway inhibitors (BRAFi ± MEKi) to target melanoma cell lines with acquired MAPKi resistance. Although similar studies have been conducted with other ERK inhibitors, GDC0994 belongs to a distinct scaffold class and differs from other ERK inhibitors by not significantly altering ERK phosphorylation levels. This may be beneficial in terms of reduced compensatory feedback activation. At the same time, GDC0994 may represent a valid option to target cancer cells which acquired resistance to another ERK inhibitor, due to its distinct binding mode as a function of its scaffold class. Therefore, we believe, that our manuscript helps to expand the current landscape of potential treatment options for malignant melanoma with or without MAPKi resistance, which should be further evaluated in future research.

Reviewer 3 Report

Comments and Suggestions for Authors

The study looks at the effect of GDC0994 as a single agent or in combination with a BRAF/MEK inhibitor in treating BRAF mutant melanoma cell lines. Even though further in vivo analysis is required to understand the real potential of GDC0994, this study provides an initial analysis of its in vitro effects, which looks promising.

Figure 1 A

The loading seems to be off, and the blot is not clear. Hence, it is not possible to confirm whether the RSK downregulation is due to differences in Loading. The same applies to the total ERK blot. These should be repeated.

Figure 2B

Please provide the detailed method of the Combination index analysis in the method section.

Figure 4.

Please add western blot data showing how the triple combination affects MEK, ERK, and RSK. Also, add MTS/MUH assay analysis of the combination.

Author Response

Reviewer 3

The study looks at the effect of GDC0994 as a single agent or in combination with a BRAF/MEK inhibitor in treating BRAF mutant melanoma cell lines. Even though further in vivo analysis is required to understand the real potential of GDC0994, this study provides an initial analysis of its in vitro effects, which looks promising.

We thank the reviewer for her/his valuable and encouraging evaluation.

Figure 1 A

The loading seems to be off, and the blot is not clear. Hence, it is not possible to confirm whether the RSK downregulation is due to differences in Loading. The same applies to the total ERK blot. These should be repeated.

We appreciate the reviewer’s observation regarding the quality and clarity of the blots. Unfortunately, we were not able to obtain clearer images within the timeframe of this revision. However, we hope that due to the striking absence of P-RSK signal as a functional correlate of reduced MAPK pathway activity in the GDC0994 and PLX4032 treated samples and the drastic reduction of P-MEK as well as P-ERK signal in the PLX4032 treated compared to the solvent control cells, the limitation posed by the somewhat unequal loading and lack of clarity of the ERK blot is negligible.

We respectfully ask the reviewer to consider this explanation in lieu of repeat experiments, and we remain committed to addressing this issue in follow-up studies or future submissions.

Figure 2B

Please provide the detailed method of the Combination index analysis in the method section.

We thank the reviewer for pointing out this crucial missing information. We have included the detailed method employed for Combination index analysis in the method section.

Figure 4.

Please add western blot data showing how the triple combination affects MEK, ERK, and RSK. Also, add MTS/MUH assay analysis of the combination.

Following the invaluable advice of the reviewer, we included Western Blot analyses to show the effect of the respective inhibitors (PLX4032, GDC0973, GDC0994) either as single agents or in combinations on the respective targets in our new Supplementary Figure S2. Additionally, MUH-based cell viability assays and analysis of potential synergistic effects of the inhibitor combinations are shown for the double resistant melanoma cell line in the new Figure 4A.

Round 2

Reviewer 1 Report

Comments and Suggestions for Authors

The authors had addressed all my concerns.

Author Response

We thank the reviewer for his comment and are happy that we could answer his questions.